# Advances Focusing on the Application of Decellularized Extracellular Matrix in Periodontal Regeneration

**DOI:** 10.3390/biom13040673

**Published:** 2023-04-14

**Authors:** Chao Liang, Li Liao, Weidong Tian

**Affiliations:** State Key Laboratory of Oral Diseases and National Clinical Research Center for Oral Diseases and Engineering Research Center of Oral Translational Medicine, Ministry of Education and National Engineering Laboratory for Oral Regenerative Medicine, West China Hospital of Stomatology, Sichuan University, Sichuan 610041, China

**Keywords:** periodontal regeneration, decellularized extracellular matrix, decellularized cell sheet, decellularized tissue, soluble decellularized extracellular matrix, guiding tissue regeneration, periodontal tissue engineering

## Abstract

The decellularized extracellular matrix (dECM) is capable of promoting stem cell proliferation, migration, adhesion, and differentiation. It is a promising biomaterial for application and clinical translation in the field of periodontal tissue engineering as it most effectively preserves the complex array of ECM components as they are in native tissue, providing ideal cues for regeneration and repair of damaged periodontal tissue. dECMs of different origins have different advantages and characteristics in promoting the regeneration of periodontal tissue. dECM can be used directly or dissolved in liquid for better flowability. Multiple ways were developed to improve the mechanical strength of dECM, such as functionalized scaffolds with cells that harvest scaffold-supported dECM through decellularization or crosslinked soluble dECM that can form injectable hydrogels for periodontal tissue repair. dECM has found recent success in many periodontal regeneration and repair therapies. This review focuses on the repairing effect of dECM in periodontal tissue engineering, with variations in cell/tissue sources, and specifically discusses the future trend of periodontal regeneration and the future role of soluble dECM in entire periodontal tissue regeneration.

## 1. Introduction

Decellularized extracellular matrix (dECM) is a promising biomaterial for repairing periodontal defects since dECM maximizes the retention of complex protein arrays, glycosaminoglycans, proteoglycans, and many other matrix components found in natural tissues [1]. Therefore, dECM provides optimal cell–ECM interactions by providing ideal biological clues to mimic native signaling events to promote the regeneration, repair, and remodeling of damaged periodontal tissue [2]. For instance, with a three-dimensional network providing a microenvironment to maintain homeostasis, support stem cell ingrowth, promote tissue formation, and initiate tissue repair, dECM has been successful in the regeneration and repair of many tissues, such as heart, nerve, and liver tissues [3]. During tissue regeneration, cell–ECM interactions coordinated with ECM component changes are vital for directing cell behaviors, functions, and fates, resulting in tissue repair and remodeling, which are regulated by specific enzymes produced by cells [3]. 

Periodontal defects caused by periodontitis, trauma, and tumors often result in the destruction of alveolar bone, periodontal membrane, and cementum. The current clinical treatment methods include basic treatment of periodontal disease, periodontal surgery, and guiding tissue regeneration, but their efficacy is often not ideal, and the orderly structure and the height of alveolar bone cannot be regenerated [1]. Several bio-scaffolds, with supportive effects, have been applied to avoid long junctional epithelium healing as a contact inhibition membrane, which is known as guided tissue regeneration (GTR). Recently, more and more biomimetic scaffolds with certain physiochemical characteristics and mechanical behavior have been applied in periodontal tissue engineering. However, periodontal tissue is a complicated tissue consisting of cementum, periodontium, and alveolar bone, which could hardly be regenerate by using a barrier membrane without providing regenerative niches with biological cues to activate the signaling pathway for the whole regeneration of cementum (C), periodontium (P), and alveolar bone (AB) simultaneously with certain structure. Moreover, the effect of biomaterials in clinical applications is usually hampered by the inertness of synthetic scaffolds compared with bioactive materials and could even result in a reverse result due to their foreignness and sometimes activate a severe immunologic reaction. 

To the contrary, natural ECM contains useful structural and biochemical information, providing sufficient bioactive cues to trigger cell functions needed for tissue regeneration. The decellularized scaffold derived from naturally occurring tissues or cultured cells maintains a natural 3D network structure and can serve as a natural scaffold. Moreover, a dECM scaffold conserves natural ECM components such as glycosaminoglycans, proteoglycans, and growth factors and could, in return, moderate immune reactions, modulate inflammatory processes, impact periodontal-related stem cell behaviors and fates, and promote periodontal regeneration. Since the complexity of periodontal tissue and tissue-specific ECM with biological cues aimed at regeneration of certain tissue (C, P, AB), it is important to provide different regenerative niches for the recruitment and modulation of endogenous stem cells for both cementum, periodontium, and alveolar bone tissue regeneration [3,4]. 

Nevertheless, advanced manufacturing techniques such as 3D printing and electrospinning have been applied to fabricate dECM-based materials/scaffolds that could mimic the unique features of periodontal tissues recently. Consequently, the dECM application form is broadened, and the clinical translation of dECM-based materials is promoted in the field of periodontal regeneration. Therefore, dECM is a vital source of bioactive material for triggering the repair of periodontal tissue and regeneration of the periodontium.

## 2. dECM Derived from Different Sources 

As a promising biomaterial, dECM has been widely applied in periodontal tissue engineering. Natural ECM generated from decellularized ECM (dECM) can be divided into two groups as decellularized cells/cell sheets (C-dECM) or decellularized tissue-specific ECM (T-dECM) according to the origins.

### 2.1. (Stem) Cell-Derived dECM (Decellularized Cell Sheet)

Significant efforts have been made to improve the quality and efficiency of the generation of cell-derived ECM with 3D structure. Part of these works involve harvesting cell-derived dECM membranes and then fabricating a 3D scaffold using various physical and chemical techniques [5]. As shown in Figure 1, C-dECM-laden scaffolds possess superior biological elasticity and porosity and allow more efficient nutrient exchange that favor cell proliferation, adhesion, and differentiation [6].

In recent years, scientists have applied certain decellularized cell sheets to periodontal regeneration. Decellularized cell sheet techniques refer to the effective removal of cellular and nuclear components of the cell sheet, especially DNA and RNA, while retaining the basic components, biological activity, and mechanical integrity of the extracellular matrix (dECM) of the sheet. Due to the lack of cells and major tissue structure, the foreign body reaction and immune rejection are significantly reduced or even absent in the cell sheet-derived dECM material, while the original three-dimensional structure, mechanical integrity, biological activity, and good biocompatibility are well preserved. Therefore, the use of decellularized sheets in regenerative medicine is increasing. 

Moreover, decellularized membrane sheets combined with cytokines or new materials to achieve better tissue regeneration effects have been used in many tissue regeneration processes, such as dental tissue regeneration [7], bone regeneration [8], vascular reconstruction [9], nerve regeneration [10], and corneal regeneration [11]. Specifically, the common C-dECM sources for periodontal regeneration are the periodontal ligament stem cell sheet (PDLSC sheet) [12], the bone marrow mesenchymal stem cell sheet (BMSC sheet) [6], the human urogenic mesenchymal stem cell sheet (hUMSC sheet), and others such as dental folicle stem cells and L929 sheet-derived dECM can also be considered.

**Figure 1 biomolecules-13-00673-f001:**
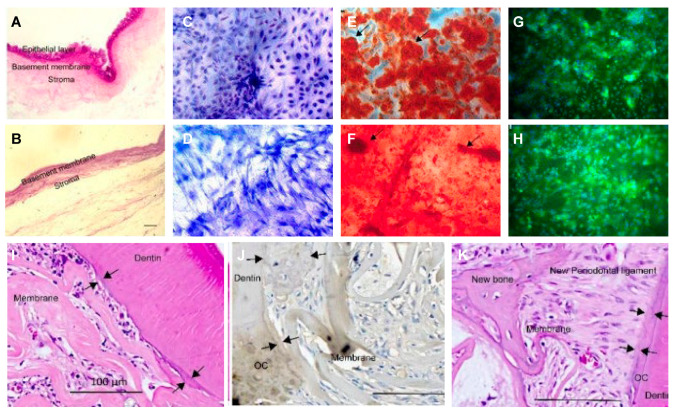
Decellularized amniotic membrane promotes osteogenic differentiation and cementum regeneration both in vitro and in vivo. Histological cross-sections of fresh (**A**) and decellularized (**B**) amniotic membrane. Representative areas of adipose-derived stromal cell cultures on polystyrene (**C**,**E**,**G**) and decellularized amniotic membrane (DAM) (**D**,**F**,**H**). ASCs confluence in the presence of supplemented medium (**C**,**D**). The osteoinduction medium stimulated the deposition of the mineralized extracellular matrix, which was observed as red calcified deposits (arrows). Osteopontin expression was evidenced by green fluorescence as agglomerates on polystyrene (**G**) and a diffuse distribution (**H**) on DAM. (**I**,**J**) and exposed old cementum (**J**,**K**), as well as a thin acellular cementum layer and thick cellular cementum (**J**). Transplantation of decellularized amniotic membrane (DAM) (**I**); DAM associated with extracellular matrix and undifferentiated adipose-derived stromal cells (**J**); scale bars of 100µm (**I–K**). The newly deposited cementum-like matrix (between arrows) is observed as a thin metachromatic adherent layer on exposed dentin Adapted from Ref. [13].

(1)Periodontal ligament (stem) cell sheets

The biological properties of human periodontal membranes were evaluated in vitro by Jiang et al. [7]. Decellularized PDLC sheets maintain an intact extracellular matrix structure, and the expression or distribution of type I collagen and fibronectin was similar in decellularized PDLC sheets and natural PDLC sheets, which contributes to the recellularization procedures and retaining the osteogenic potential of allogeneic human periodontal ligament stem cells. Four weeks after surgery, newly formed bone, cementum, and periodontal membranes were observed with a decellularized PDLC sheet combined with 15-Deoxy-Δ (12,14)-prostaglandin J (2) nanoparticles. Thus, it is believed that hPDLC sheets may be a potential extracellular matrix source for periodontal regeneration in the oral inflammatory environment. In addition, other results also indicate that decellularized sheets derived from PDLSC can improve the proliferation and differentiation of mesenchymal stem cells, thus performing regenerative potential in the future of periodontal tissue engineering [12,14]. Moreover, PDLSC sheet-derived dECM is believed to preserve growth factors, such as fibroblast growth factor (bFGF), vascular endothelial growth factor (VEGF), and hepatocyte growth factor (HGF) [14].

(2)BMC/BMSC sheet:

The 14-day BMSC sheets cultured in vitro significantly promoted osteogenesis of bone marrow stem cells compared with counterparts by Xu et al. Proteomic analysis showed that the extracellular matrix components of the cell sheet varied over time. Type IV collagen chains in 14-day dECM were higher than those in other membranes and more able to promote osteogenic differentiation of BMSCs [15]. Type IV collagen was significantly enriched in the activated focal adhesion kinase/phosphatidylinositol-4,5-diphosphate 3-kinase/protein kinase B signaling pathway (FAK/PI3K/AKT). Thus, 14-day dECM with a high content of type IV collagen chain enrichment that could promote osteogenic differentiation of BMSCs may be related to the activation of the FAK/PI3K/AKT signaling pathways [4].

(3)Urogenic mesenchymal (stem) cell sheets:

Yang et al. found that the conditioned medium of hUMSC contained components that could promote cementum differentiation [16]. Xiong et al. seeded human UMSCs on plates and induced the extracellular matrix generation, followed by decellularized processing, leaving the extracellular matrix components on the dishes. Using decellularized hPDLSC sheets and fibronectin-coated plates as controls, hUMSCs had more fibronectin components than hPDLSCs, which better promoted the proliferation, adhesion, and osteogenic differentiation of hPDLSCs [17].

Many studies have raised the question of whether dECM derived from different cell types has different effects on cell growth. Hoshiba et al. studied dECM extracted chemically or enzymatically from chondrocytes produced from chondrocytes, fibroblasts, and bone marrow mesenchymal stem cells and investigated their effects on articular chondrocyte behavior [18]. ECM derived from chemical methods is significantly superior to enzymatic methods in cell adhesion, and the number of chondrocytes adhering to chondrocyte-derived ECM is significantly higher than that of fibroblast or mesenchymal stem cell-derived ECM. ECM generated from the same cell lineage as the target tissue has been shown to provide a more appropriate microenvironment for cell growth because of unique molecules and pores of appropriate size, which also promote cell reinoculation and cell adhesion [18].

The C-dECM, with several unique benefits, has attracted considerable interest (Table 1). C-dECMs are usually harvested after decellularization of cell sheets constructed in vitro for a few weeks. The cell-sheet technique allows target cells to reproduce in large amounts and harvest enough C-dECM for clinical use since ECM is the major component of the cell sheet while cells, especially stem cells, could proliferate indefinitely. Acquiring C-dECM, especially for obtaining dECM from certain progenitors or stem cells, is relatively simpler when compared to the sophisticated fabrication of tissue-like dECM [19]. Moreover, C-dECMs have an advantage of safety since in vitro cell expansion can reduce the risk of pathogen transfer caused by allogenic ECM and avoid adverse host immune responses induced by xenogeneic ECM. Furthermore, in-vitro cultured cell-derived dECM can be more operable to modify, such as when cocultured with osteogenic inducers to harvest mineralized C-dECM or to graft on other surfaces of biomaterials [20], such as treated dentin matrix, hydroxyapatite, and biphasic calcium phosphate, to form another excellent periodontal tissue engineering construct [21].

Although C-dECMs could help to provide a microenvironment with the help of omplicated manufacturing for relative tissue regeneration, reconstructing the natural 3D architecture of the whole organ or tissue in the laboratory is now unrealistic [22]. 

Therefore, C-dECM couldn’t perfectly mimic the sophisticated ECM structure and interior architecture, such as the porous structure and the collagen fiber arrangement, which might have an effect on cell adhesion, proliferation, and differentiation [23]. Furthermore, the mechanical properties and microenvironment conditions of cell-deposited ECM are different from those of native ECM. Unlike allogenic T-dECMs serving as reservoirs for site-specific bioactive molecules, C-dECM was believed to be inferior in preserving tissue niches for tissue regeneration when compared with T-dECM [12,24,25].

**Table 1 biomolecules-13-00673-t001:** Cell-derived dECM for periodontal regeneration.

dECM Origin	Cell/Cytokine/Scaffold	Model	Outcome
PDLC sheet	15-Deoxy-Δ (12,14)-prostaglandin J (2) nanoparticles/nanofiber scaffold	Rat periodontal defect model	Four weeks after surgery, newly formed bone, cementum, and periodontal membrane were observed [7].
PDLSC sheet	PDLSC	In vitro	Promote periodontal membrane differentiation and osteogenic differentiation [14].
PDLSC sheet	PCL scaffolds	In vitro	More new attachment formation was observed [12].
UMSC sheet	PDLSC	In vitro	Promote cementogenic differentiation [16].
UMSC sheet and PDLSC sheet	PDLSC	In vitro	Both UECM and PECM promoted hPDLSC proliferation, attachment, spreading, and differentiation. UECM showed advantages in enhancing proliferation, osteogenesis, and angiogenesis [17].
BMSC sheet	BMSC	Rat calvarial critical size defect model	Fourteen days of dECM with a high content of type IV collagen chain enrichment can promote the osteogenic differentiation of bone marrow mesenchymal cells by the FAK/PI3K/AKT pathway [4].
BMSC sheet	BMSC/Electrospun poly (e-caprolactone) (PCL) fifiber mesh scaffolds	In vitro	ECM-containing constructs maintained osteogenic differentiation ofMSCs [26].
MC3T3-E1 cell sheet	BMSCs/GelMA	critical-sized segmental bone defects at rabbit radius	pODM can be effectively used as a biomimetic periosteum that preserves the desirable biological molecules of ECM, but without the concerns of using exogenous cells [27].
BMSCs, MC3T3 osteoblasts, L929 fibroblasts	BMSCs/electrospun PLLA/gelatin fibrous meshes	C57BL mice Subcutaneous implantation	Different celltypes produce D-ECMs with different amounts of collagen, GAG and bioactive factors because of the differences in cell proliferation and differential potential. For the osteogenic differentiation of multipotential BMSCs, it was identified that dECM derived from cells originated from bone tissues showed the strongest promotion of the osteogenic differentiation of reseeded BMSCs [22].

### 2.2. Tissue-Derived Extracellular Matrix for Periodontal Tissue Engineering Constructs

Despite the similar ECM composition between different tissues and organs, subtle differences in function, proportion, structure, and stiffness of the ECM can influence the interaction of cells in determining cell fate (Figure 2). Tissue dECM can promote tissue-specific and non-tissue-specific stem cells/progenitor cells, primary cell proliferation, and serve as a tissue-specific scaffold for stem cells/progenitor cells. Even without a specific differentiation mediator, the stem cells or progenitor cells still have the corresponding cell lineage differentiation capacity, based on the specific interaction between the cell and the ECM [28]. Thus, tissue dECM has an advantageous role [28] in maintaining and directing stem cell differentiation [26] (Table 2). 

#### 2.2.1. Human (Allogenic) Tissue Derived dECMs

(1)Dental (craniofacial)-related human tissues derived dECM

To achieve the goal of complete periodontal regeneration, many studies have evaluated the effectiveness of decellularized tissue derived from human tooth-related tissue for tissue engineering. Hyoju Son et al. [30] applied sodium dodecyl sulfate and Triton X-100 separately to treat tooth slices harvested from healthy human third molars, which was verified to efficiently remove nuclear components and maintain intact structure and composition of dECM. Furthermore, T-dECMs can support the regeneration of PDLSCs in the vicinity of cementum and express cementum-related and PDL-related genes. These results indicated that the dentin dECM promoted proliferation and differentiation of MSCs, thus having regenerative potential in future tissue engineering for periodontal regeneration. 

The human tooth dECM has also been demonstrated to promote the differentiation of PDLSC, as proved by Ik-Hwan Kim et al. [31]. The cementum/PDL-anchored structures can be regenerated after PDLSC migration as expected, which suggests that decellularized human tooth material may serve as a biological scaffold for PDLSC loading as a novel therapeutic approach for periodontal defects.

As for the complicated composition of different human-derived tissues dECM and the related functions, they vary from type to type. Dental tissue engineering approaches make it easier to solve this problem with dECM constructs generated from tissues derived from the same origin as extraembryonic mesoderm, which are believed to have more bioactive molecules naturally. Dental-derived tissue dECM has been widely used to promote dental tissue repair and has achieved great success [32,33,34].

(2)Non-dental-related tissue-derived dECM

a.Human amnion dECM

Decellularized human amnion has been adopted as a soft tissue replacement and a delivery system for stem cells. Dilcele Silva Moreira Dziedzic et al. applied the decellularized amnion infiltrated by adipose mesenchymal stem cells (ADSCs) with mineralized extracellular matrix to a rat periodontal root bifurcation lesion model [13]. After being treated with the decellularized amnion/ADSCs/mineralized ECM, periodontal healing was assessed by micro-CT and histological analysis when compared with untreated defects. Decellularized amniotic membranes with or without ADSCs promoted bone healing compared with controls. What’s more, it has also been observed that human-derived decellularized amniotic membranes enhanced periodontal tissue formation in PDLSC defects [35]. The effects of human decellularized amnion on periodontal regeneration were widely explored and showed great clinical translational potential, as a number of randomized controlled treatments proved [36,37,38,39]. 

b.Human umbilical vein dECM

Allogenic “scaffolds” derived from the human umbilical veins (HUVs) were fabricated, and the biological and mechanical properties during early remodeling events were examined. Human umbilical vein-derived dECM harvested by the osmotic lysis method displayed increased cellular proliferation and reduced metabolic activity compared to HUVs treated with surfactant. Biomechanical properties were largely preserved and similar when subjected to different treatments, and the results suggest that by optimizing processing conditions, biological events associated with remodeling can be modulated to tailor function for specific clinical applications. 3D dECM derived from the HUVs has potential for functioning as a regenerative matrix for tissue regeneration with ideal biomechanical properties [40]. Allogenic dECM derived from the human umbilical vein [41] were proved to have ideal biological behavior in promoting proliferation and reducing metabolic activity in mesenchymal-derived cells. The specific structure of the vein has the lumenal surface acting as a barrier to inhibit cell migration, while the ablumenal surface promotes cellular invasion at the interface with the wound site [40].

#### 2.2.2. Heterogenous Tissue-Derived dECMs

(1)Heterogenous dental (craniofacial)-related tissues derived from dECM

a.Decellularized porcine dental matrix

Han et al. [42] treated the incisor teeth with the decellularized method to obtain the porcine dental dECM, combined the dECM with the immunomodulator RSG, and evaluated the physical and chemical characteristics of the extracellular matrix and its effects on tissue regeneration. The results showed that the extracellular matrix ensured the proliferation and differentiation of odontogenic stem cells with good immune regulation, antagonizing the classical activation of M1 macrophages. Selective activation of M2 macrophages promoted the regeneration of the ligament-bone interface. At the same time, the tissue-engineered scaffold significantly reduces the absorption of bone around the implant and expresses the protein markers of natural cementum and alveolar bone.

b.Decellularized matrix of dog periodontal ligament

The effect of dog periodontal acellular matrix on periodontal healing was evaluated by Lee et al. [43]. The mandibular premolars from six beagles were either untreated, decellularized, or treated with root planning before being reimplanted into the extraction site. A chemistry method could satisfy the need for PDL decellularization, as confirmed by cytological, histological, SEM, and TEM analysis. After 8 weeks, both the untreated PDL group and the decellularized PDL group supported periodontal reattachment (especially cementum formation) to some extent at the extraction site. More periodontal regeneration was seen in the decellularized periodontal tissue group than in the teeth in the root planning group.

c.Decellularized rat mandible matrix

Naoko Nakamura et al. [44] prepared rat mandible matrix with high pressure perfusion, DNA enzyme, and detergent and evaluated the reconstruction of periodontal structure as periodontal matrix in vivo. After tooth extraction, the decellularized mandible with periodontal matrix was implanted into a rat kidney capsule, and the host cells were observed to migrate into the periodontal matrix and align along the collagen fibers of the periodontal membrane, demonstrating that the decellularized mandibular matrix can reconstruct periodontal tissue by controlling host cell migration and can be used as a novel periodontal therapy.

(2)Heterogenous, non-dental-related tissue-derived dECM

Solid dECM from non-dental-related tissue sources has recently been studied for periodontal defect regeneration due to its rich resources, economic benefit, massive production, and usually harvesting from discarded tissue, such as treated porcine small intestinal submucosa (SIS) and the extracellular matrix of amnion and pericardium.

a.SIS 

SIS is one of the most commonly used decellularized tissues, with good potential for soft tissue repair and regeneration, but it lacks sufficient mechanical properties to guide bone tissue regeneration unless properly modified. Gu et al. used the pig small intestinal submucosal tissue as a biological collagen membrane for guiding tissue regeneration [45]. In the rat cranial defect model, the SIS dECM could promote bone tissue regeneration, had a higher bone volume fraction, and regenerated more bone tissue, presenting the potential to promote bone regeneration in periodontal defects.

b.Decellularized Amnion

Decellularized Amnion is of great biocompatibility and can maintain original tissue structure, support cell attachment, cell infiltration, bone deposition, and periodontal regeneration [46]. The amniotic membrane from various species were used to regenerate craniofacial bone by a few scientists [47]. The amniotic membrane contains collagen type I, which helps improve the formation and strength of the bone, and this instinctive property makes it ideal for tissue regeneration procedures.

c.Decellularized pericardium

Serena Bianchi et al. introduced the decellularized bovine pericardium as a membrane material loading PDLSCs [48]. The decellularized bovine pericardiums retained the three-dimensional collagen structure for stem cell infiltration. hPDL fibroblasts were seeded on the decellularized bovine pericardium and were healthy, large, and polygonal with filopodia and lamellipodia. Other results show that cells migrate along and within the membrane layer, binding to membrane fibers by filamentous extension, showing that decellularized bovine pericardium could serve as a carrier for PDL fibroblasts [48]. 

The porcine pericardium [49,50] is also a commonly used material for periodontal ligament regeneration, and Mika Suzuki et al. [41] verified the important role of the decellularized pericardium in promoting periodontal ligament structural remodeling and promoted the gradient mineralization of the decellularized pericardial surface to promote the formation of ligamento-structure at the soft and hard tissue interface. Rabbit pericardial tissue [47] has been used to restore craniofacial bone defects, and the decellularized rabbit pericardial was believed to facilitate bone regeneration by inhibiting connective tissue invasion.

d.To sum up, decellularization of natural tissues to produce extracellular matrix is a promising method for 3D scaffolding and for investigating cell-ECM interaction during regeneration of target tissue [3,51]. The fate and behavior of mesenchymal stem cells are influenced by the stem cell niches ideal biochemical and physical cues. Ana Rita Pereira et al. compared the biological behaviors of BMSCs when exposed to C-dECM and T-dECM, and better outcomes were observed in 3D decellularized bone tissue for greater architecture complexity and physicochemical properties [52]. To sum up, tissue-derived dECM has great potential in the context of endogenous periodontal regeneration, with a better effect on preserving the tissue niche intended for different tissues in periodontal defects.

**Table 2 biomolecules-13-00673-t002:** Tissue-derived dECM for periodontal regeneration.

dECM Type	dECMOrigins	Host Species	Cell/Cytokines	Scaffold	Model	Outcome
Dental (cranial)-associated tissue derived-dECM	Swine deciduous incisor teeth dECM	Immunodeficien mice	hDFCs/rat DFCs/RSG	Cone shape scaffold, 10 mm in length, 4 mm in diameter	Dorsum transplanted in immunodeficien mice	Effectively decrease the expression of IL-1 and TNFα, and increase the expression of IL-10 and TGFβ to enable favorable immunomodulation and promote the soft/hard interface and their effective integration with host-local tissue by PPAR γ to induce alternatively activated macrophages [42].
	Mouse decellularized mandible bone with a PDL matrix	Rat	-	-	Implanted under the subrenal capsule in rat	Decellularized PDL matrix retained the collagen fiber structure and can reconstruct PDL tissue by controlling host cell migration, which could serve as a novel periodontal treatment approach [44].
	Decellularized human tooth slices	-	PDLSCs	-	In vitro	A decellularized scaffold could support repopulation of PDL stem cells near the cementum and express cementum- and periodontal-ligament-related genes [30].
	Decellularized Human Tooth Scaffold	Immunosuppressed mouse	PDLSCs DPSCs		Subcutaneously transplantation	A regeneration of the cementum/PDL complex could be expected [31].
	Porcine SIS dECM	Rat	PDLSCs	-	Rat cranial defect model	dECM promotes bone tissue regeneration, and the crosslinked dECM has a better effect on regenerating more bone tissue [45].
Non-dental(cranial)-associated tissue derived-dECM	Human Decellularized Amniotic Membrane	Rat	Adipose-derived stromal cells	Mineralized extracellular matrix	Rat periodontal furcation defect model	DAM promoted neovascularization and promoted osteoconduction DAM with ASC or without cells, and the ECM ensures bone tissue healing [13].
	Bovine Pericardium Membranes	-	Human periodontal ligament fibroblasts	-	In Vitro	Cellular migration along and within the layers of the membrane, binding with membrane fibers by means of filopodial extensions [48].
	Porcine decellularized pericardial tissues	-	NIH3T3, C2C12, and mesenchymal stem cells	-	In Vitro	3D-reconstructed decellularized pericardium with cells has the potential to be an attractive alternative to living tissues, such as ligament and tendon tissues [49].
	Epiphyseal plate	Porcine	-	-	hBMSC	Support hBMSC proliferation and an increase in the expression of collagen types I, II, and X [8].

## 3. Application of dECM of Different Forms in Periodontal Regeneration

Currently, there are many studies using different types of dECM for periodontal treatment, including solid-state dECM for the treatment of periodontal defects and soluble dECM-derived materials such as injectable dECM hydrogels. Solid-state dECM has the advantage of accessibility, mechanical strength, preservation of ECM structure and components, and ease of handling compared with soluble dECM. While soluble dECM material is easier to modify into different shapes adapted to the defect area, such as through 3D printing, and could be used in minimal injury treatment such as hydrogel injection. Previous studies used to focus on the application of solid-state C-dECM and T-dECM, and solid tissue has a natural advantage in the recellularization of cell seeds with more well-preserved dECM components, while dECM bioink with ideal flowability and viscosity makes it applicable to encapsulate seed cells directly. Therefore, the adoption of dECM in different states of periodontal tissue depends on the purpose and characteristics needed.

Solid-state dECM functioned as a “scaffold” that underwent decellularization and was used directly as a biomaterial without further disrupting the dECM microstructure. Solid-state dECMs can be classified according to the mode of application. These include acellular membrane sheets, dECM patches/sheets, and the whole periodontal tissue. Soluble dECM is a material that has been decellularized; it then takes additional steps to resolve the extracellular matrix structure and dissolve the dECM into a liquid form. Soluble materials can be classified according to recombinant or application methods, including powders, microdroplets, injectable hydrogels, 2D and 3D hydrogels, and periodontal regeneration modules consisting of soluble dECM and other biomaterials. This review will focus on the different types of dECM in the context of periodontal regenerative applications with tissue and species sources and discuss in detail advances in the development, in vitro studies, in vivo implementation, and clinical translation of solid and soluble dECM.

### 3.1. Solid Decellularized Extracellular Matrix (s-dECM)

This section focuses on solid-state dECM stents, which do not dissolve into dECM powder or liquid and are used directly after decellularization. This approach preserves the native tissue architecture in the process of expressing the dECM component at its tissue-specific location. Therapies using this solid decellularization approach focus on either directly using the material as a solid-state module or membrane material or on trying to recellularize the dECM and develop into functionalized tissue (Figure 3).

To evaluate the effects of dECMs with different physiochemical characteristics and biological components on cell behavior and tissue repair in periodontal regeneration, many researchers have applied dECMs to promote the regeneration of periodontal tissue. In addition to the direct application of the above cell- and tissue-derived ECM, it is common to combine solid decellularized extracellular matrix with biomaterials, chemicals, cytokines, and other biological cues to improve their physicochemical properties. dECM has been constructed in vitro or harvested from decellularized tissue, and it has been combined with various modification methods to establish tissue engineering constructs that meet the needs for periodontal regeneration.

(1)Direct application of pure dECM

Cell- and tissue-derived dECM, with the ideal composition, structure, and mechanical properties of the extracellular matrix after cellular components are removed, is usually in a sheet-shape or block mass and has been directly used to fill the periodontal defect area in the application practices of periodontium reconstruction, cementum regeneration, and bone tissue engineering.

Cell-derived dECM could be harvested from rich cell types and could easily get a great amount of dECM satisfying the application need. Pre-osteoblast (MC3T3-E1 cell sheet)-derived matrix was found to exhibit considerable chemotactic effect and osteogenic induction capability to bone marrow mesenchymal stem cells (BMSCs) [27]. Moreover, different types of C-dECMs have been verified for their capacity to induce osteogenic differentiation of re-seeded BMSCs, including cell sheets derived from bone marrow mesenchymal stem cells (BMSCs), MC3T3 osteoblasts, and L929 fibroblasts [22]. In addition, C-dECM showed good effect in ligament regeneration: stem cell sheets-derived dECM provide an ectogenic scaffold-free option, enhancing bone formation and angiogenesis by BMP-2 and VEGF, modulating macrophage polarization and MMP/TIMP expression, and physically promoting ligament reconstruction [55]. In addition, many techniques have been used to construct C-dECM in vitro; for example, studies showed that the addition of a certain amount of ascorbic acid to the culture medium promoted ECM deposition and the formation of cell membranes [56].

In addition, the further treatment of decellularized tissue to make it meet the application needs of periodontal tissue engineering is also a problem that scientists try to solve. The most commonly used decellularized tissue in periodontal regeneration is decellularized bone [57,58]. Bone ECM showed an inductive effect on the production of new bone by osteoblast-lineage cells [58]. In addition, decellularized allogeneic bones with the Haversian System contribute to angiogenesis [57]. For example, vascularized, decellularized bone combined with a vascular pedicle proved to have a promising effect on bone regeneration, representing potential therapeutic alternatives for bone tissue-free transfer in periodontal bone defects [59]. Moreover, book-shaped acellular fibrocartilage enlarges seeding cells loading, showing superior inducibility to stimulate collagen or glycosaminoglycan secretion in vitro and good interface integration in vivo [60]. Decellularized SIS was crosslinked to enhance its mechanical strength with EGCG, a natural cross-linking agent with osteogenic activity [45].

Other shapes of 3D cell-derived dECMs with spatial structure that depend on polymers such as poly (lactic-hydroxylactic acid) (PLGA) and polylactic acid (PLA) as expendable templates have also been developed. In these cases, cells are directly seeded onto the template surface, which in turn stimulates ECM deposition to form a cell-ECM-template structure. Decellularized methods are being used, and the polymer template will be removed by chemical reagents, leaving only cell-derived dECM scaffolds with three-dimensional shapes. With a spatially structured cell-derived dECM-laden scaffold, exhibiting a mild host immune response and good biocompatibility when in vivo.

(2)dECM-based hybrid scaffold

Similar to three-dimensional cell-derived dECM scaffolds with specific spatial structures, a novel cell-derived ECM hybrid scaffold was generated by covering the template surface with cell-derived ECM and then retaining the template, making these templates biologically active. The surface bioactivity of the mixed scaffolds provides greater control of cell fate for more beneficial interactions between cells and biomaterials. Furthermore, cell-derived ECM hybrid scaffolds with stronger mechanical properties than tissue or cell-derived ECM products lessen the applications of dECM quantity and improve efficacy in tissue engineering, particularly those associated with bone tissue engineering, which are believed to have a strong correlation of extracellular stiffness.

Farag et al. constructed tissue-engineered composite constructs by loading PDLSCs on a polycaprolactone scaffold and then decellularizing them. The effects of tissue-engineered constructs on differentiation of allogeneic PDLSCs and other MSCs were assessed by gene expression of bone markers and periodontal ligament markers after cell seeding, confirming that the decellularized hybrid tissue-engineered constructs were able to induce cell differentiation in vitro [12]. Polycaprolactone/gelatin nanofibers were prepared as a carrier to improve the mechanical strength [7] of human periodontal membrane cell membrane dECM by Jiang et al. Besides, Yu et al. applied preosteoblast (MC3T3-E1 cell sheet)-derived matrix as a biomimetic periosteum supported by gelatin methacryloyl hydrogel (GelMA) to promote bone regeneration [27].

Gu et al. modified the porcine SIS by EGCG crosslinking, which increased the SIS to four layers, greatly improving its mechanical strength. The EGCG crosslinking significantly improved the mechanical properties and hydrophilicity of the submucosa of decellularized SIS while maintaining good cell affinity. In contrast to decellularized pig small intestinal submucosa, EGCG cross-linked decellularized pig small intestinal submucosa enhances adhesion of fibroblasts and pre-osteoblast cells and promotes osteogenic differentiation of MC3T3-E1 cells. In the rat cranial defect model, the crosslinked material had a better occlusion effect and accelerated bone regeneration, acting as a promising guiding tissue regeneration membrane material [7].

At present, there are many studies to modify dECM-based materials combined with other cytokines, immune regulators, anti-inflammatory components, and antibacterial materials to construct a tissue-engineered periodontal regeneration module. For example, deoxyprostaglandin nanoparticles were found to be anti-inflammatory and improve regeneration by Jiang et al. [7]. Cytokines such as BMP-2, FGF, VEGF, and TGF-β were normally added in periodontal regeneration with positive results, and the combination of cytokines and other bioactive extracts with dECM materials achieved a greater outcome in both bone tissue engineering and periodontium regeneration [57,58,59,61,62].

Recellularization/stem cell homing promoted or mediated by dECM-based materials count for a great deal in periodontal tissue repair. On the one hand, dECM with biological cues serves as a tissue engineering construct that could promote stem cell homing. On the other hand, dECM-based material serves as a vehicle laden with a great amount of tailored stem cells to provide the defect area with a greater amount of stem cells and a more active cell-ECM reaction in periodontal tissue regeneration in the first place [32]. For example, PDLSC seeded on an acellular amniotic membrane promoted the repair of periodontal ligaments, cementum, and alveolar bone and induced periodontal regeneration in rats [35,63]. Moreover, DPSC combined with decellularized bone Bio-Oss^®^ supported by an acellular amniotic membrane induced more CM and PDL than the polymer scaffold group, according to Khorsand et al. [64].

To sum up, there has been established a shared set of rules for solid state decellularized material in periodontology. The methodology of dECM materials relies on application need and has seen great success in promoting periodontal tissue repair and reconstruction. However, with the development of advanced manufacturing techniques, more and more scientists focus on the integration (to avoid shrinkage, collapse, and deformity), cell-laden ability (flowability and solubility), and minimally invasive effect (injectable and phase change) of tissue engineering constructs for periodontal regeneration. Therefore, some scientists found a solution to improving the properties of dECM with the invention of soluble dECM.

### 3.2. Soluble Decellularized Extracellular Matrix

This section focuses on soluble dECMs, which prepares solid dECM into dECM powder or dissolved into a liquid. This approach both retains the natural ECM structure and increases its flowability. Treatments using this dECM method for periodontal tissue engineering mainly include minimally invasive hydrogel injection therapy, light-curing hydrogel synthesis, and 3D-printed bioink manufacturing.

#### 3.2.1. Development of Soluble dECM

The lysis method is based on porcine SIS dECM, which was first developed by Freytes et al. in 2008 and shortly followed by dissolved dECM [37,38] by Singelyn et al. in 2009. Currently, multiple methods have been developed to generate a soluble dECM. Soluble dECM has been obtained from various animal sources, including the pulp, heart, liver, esophagus, small intestine submucosa, placenta, and omentum [54]. Organs and tissues are also decellularized using a similar protocol, among which the most commonly used is the chemical/detergent method. Typically, organs are cut into small pieces and merged in solution, which is then lyophilized and ground into a fine powder, followed by pepsin digestion with low concentrations of pepsin for several days to further decompose the dECM and improve the solubility. The pH of the solution was adjusted to be alkaline to inactivate pepsin, and the salt concentration of the solution was then adjusted. Soluble dECM self-assembled into a nanofiber hydrogel after 30 min to 1 h at physiological temperature.

Soluble dECM is usually used in dental-derived dECM, such as porcine pulp dECM, dog periodontal membrane dECM, and even human dental capsule dECM. However, non-dental dECM, such as porcine SIS, acellular amnion, etc., needs to increase additional biological signals to promote periodontal regeneration, such as fibroblast growth factor [65], platelet fibrin [66], and other cytokines and mesenchymal stem cells, such as those related to periodontal regeneration, such as PDLSCs, BMSCs, DFSCs, ADSCs, and others [13].

#### 3.2.2. Application Practice of Soluble dECM

(1)Grinded dECM powder/granule

Grinded and partially dissolved dECM has been used directly for tissue engineering applications. This may be due to the fact that granular dECM could contain more repair components than fully dissolved dECM. Grinded dECM powder can be partially dissolved in solution or resuspended to form a suspension for the injection without complete dissolution [67,68], which may retain the structure and protein composition of ECM better than soluble ECM while the suspension has fluidity that can be injected into the defect [69].

These methods rely on direct injection of partially dissolved dECM powder or dECM microparticles. For example, partially digested adipose tissue dECM powder has been used as an injectable material for adipose tissue engineering. In in vitro studies, dECM powder can very effectively promote cell attachment and proliferation [13]. Thus, human dECM powder can be used as efficient injection biomaterials for tissue engineering with great potential in addressing the clinical challenges of periodontal regenerative medicine [8,70,71]. In addition, decellularized bone powder, as a commonly used periodontal tissue engineering material, is another form of clinical application of dECM granules, which are often combined with hydrogel and a guiding tissue regeneration membrane to promote periodontal regeneration [50,71,72]. For example, Sayanti Datta et al. applied decellularized bone granule material incorporated in a chitosan-base hydrogel laden with hAMSC to promote bone regeneration, and intrinsic growth factors in the decellularized bone granules encouraged excellent cellular viability, proliferation, and osteogenic differentiation in human amnion-derived stem cells [72].

(2)Nanoparticles/microdroplets

Grinding particles vary greatly in size and cannot be fully dissolved, which affects the homogeneity of dECM materials and impacts the therapeutic effect. To obtain dECM particles with more bioactive characteristics as nanomaterials while preserving the regenerative effect, the dECM solution could be electrosprayed into nanoparticles. A method for homogeneous dECM nanoparticles was developed by Link et al. [73]. The group first solubilized dECM with acetic acid and then electrospray dECM droplets onto aluminum foil and collected them with a smaller size distribution of 225 ± 67 nm and a higher zeta potential (10 ± 1.6 mV), The dECM particles/powder produced by electrospray have a natural fiber-like structure, contain various ECM proteins, and are easily resuspended in the buffer and gelatin solution [74]. Farzane Sivandzade et al. adopted crosslinking techniques to obtain injectable microcarriers with fibrous structure by creating an enzymatically digested ECM solution mixed with a Chitosan solution containing crosslinking agents, which was then dropped into a crosslinker solution through a 16-gauge needle driven by a syringe pump under a high-voltage electrostatic field to produce spherical-shaped microcarriers [75]. The microdroplets have key characteristics, including appropriate mechanical strength, porous microstructure, and pore size, as well as controlled biodegradability and biocompatibility, and are therefore promising in skeletal tissue engineering.

(3)Hydrogel

ECM-derived hydrogels remove cellular components and DNA, retain sGAG and other ECM proteins, such as collagen, gel at physiological temperature and pH, and have nanofiber structure [76].

a.2D coating and hydrogel

The platform of dECM-based scaffolds via 2D coating or hydrogel systems provides an excellent method to assess cellular responses to different dECM, allowing high-throughput analysis and evaluation of rare or inaccessible pathological parameters of dECM [77,78]. Tissue-specific dECM can be used to coat cell culture dishes in ways similar to other materials such as collagen used for dish coating or can polymerize into thin hydrogels for cell culture on tissue engineering constructs, especially in the culture and expansion of stem cells as ECM could better preserve the stemness of stem cells [17,28,79,80,81]. Moreover, ECM was found to mitigate the chronic inflammatory response and avoid the detrimental human immune response to metal implants or synthetic implants [82,83]. These soluble dECM platforms have also been used to mimic the tissue in vitro microenvironment and can also be used to assess the ability of dECM to regulate stem cell biological behavior and phenotype in vitro. Simon Farnebo et al. [84] synthesized a biocompatible hydrogel composed of tendon dECM from human tendon that was described as a potential bioscaffold for guiding tissue regeneration and tissue engineering purposes. Wang et al. detected higher levels of OCN and Nanog in BMSC when seeded on ECM-coated titanium substrates and showed a positive effect of cell sheet-derived ECM coatings on the proliferation and differentiation of BMSCs, which play an important role in bone tissue engineering and could substantially promote periodontal regeneration [80]. Weigel et al. have proved that biomimetic scaffolds coated with extracellular matrix (ECM) synthesized by human fibroblasts demonstrated improved cell adhesion [81].

Here, we expect that the use of extracellular matrix hydrogels derived from decellularized tissues/cells can provide an environment capable of directing cell growth. These gels have biochemical features of the tissue-specific extracellular matrix and have the potential for clinical translation. For example, hydrogels from the porcine SIS dECM can enable the formation and growth of endoderm-derived human organs, such as the stomach, liver, and pancreas [24,54,70]. Furthermore, dECM gels can be used as direct human organoids with cell growth and stable tissue hierarchy features for tissue engineering and drug selection. The development of these dECM-derived hydrogels opens up the potential for tissue engineering [54]. In addition, there are hybrid hydrogels with improved physicochemical properties derived from dECM hydrogels that combine with other traditional hydrogels with specific properties. A mixed hydrogel consisting of porcine corneal stroma dECM and methacrylate hyaluronic acid was developed by a noncompetitive double-crosslinking process by Shen et al. [85]. Hybrid hydrogel not only preserved the biologically active components of the extracellular matrix but also matched corneal transparency with a low swelling ratio. dECM hydrogels were improved with slow degradation and enhanced mechanical properties by hyaluronic acid function groups and with high adhesion characteristics because of dECM proteins, effectively promoting tissue regeneration after corneal defects [85].

b.3D hydrogel

In clinical practice, cells are exposed to a 3D environment. Hydrogels with a specific spatial structure may significantly alter the cell properties and the repairing potential compared to a 2D model. In the application of 3D bio-printed tissue engineering modules to promote periodontal regeneration, compared to purely injectable dECM materials, preparing spatially-stable dECM hydrogel is challenging due to its soft mechanical properties and process difficulties. 3D printing technology is a useful method, and Seung Hyeon Hwang et al. prepared scaffolds with a three-layer dECM hydrogel patch with spatial segmentation, providing controlled release of multiple growth factors and inducing good neovascularization for tissue regeneration [86].

Extruded 3D bioprinting relies on structurally supported bioink layers stacking on each other, requiring high-viscosity solutions for proper printing and scaffold fabrication [78]. At the concentrations commonly used for dECM injection, the dECM prepolymers cannot form layers or print due to their very low viscosity. Three-dimensional bioprinting technologies such as extruded bioprinting have attempted to address these issues with higher dECM concentrations, crosslinking modifications, and incorporation with other biomaterials.

Moreover, 3D hydrogel could support 3D printing of periodontal regeneration constructs [78]. Yang et al. [87] applied dissolved dECM as cell-laden bioink, combined it with methacrylate gelatin to increase the printability of the bioink, and designed a 3D bioprinted, light-cured biomimetic periodontal tissue engineering module. It is confirmed that dECM has potent immunomodulatory activity, reducing proinflammatory factors released by M1 macrophages and reducing local inflammation in periodontal defects in SD rats. In the clinically relevant critical-size periodontal defect model, the bio-printed module significantly enhanced the regeneration of periodontal tissue, especially forming the anchored structure on the ideal bone-periodontal interface with well-arranged periodontal fibers and highly mineralized alveolar bone. This demonstrates the effectiveness and feasibility of 3D bioprinting combined with dental follicle-specific decorative dECM bioinks for periodontal tissue regeneration, providing new avenues for future clinical practice. In another study, a 3D-printed ECM-based scaffold system using vitamin B2 UV-induced crosslinking without support material was developed by Jang et al. [88]. Nevertheless, Cairong Li et al. developed a biocompatible bioink (ECM @ MeHA bioink) with suitable mechanical support and visible light printable properties, which inspired the application of a hybrid hydrogel consisting of both ECM and non-ECM material in the field of periodontal regeneration [89].

Therefore, 3D ECM hydrogel has been viewed as a customized scaffold with promising characteristics [90] for three-dimensional (3D) cell cultures to promote osteogenic differentiation, combinations with other biomaterials to increase their osteogenic effects, and mimicking the in vivo microenvironment to promote bone repair [90,91]. Recent emphasis on 3D applications of ECM lies in three-dimensional bioinks based on ECM for biomedical applications, especially defect-tailored 3D printing.

Entire periodontal tissue engineering has been a shared goal of scientists aiming for periodontal regeneration. However, most of the research addressing the periodontal defect focuses on alveolar bone regeneration [92], even for clinical translation. Since bone regeneration with a shallow periodontal pocket could provide adhesive sites for periodontium fibers and help limit the recession of soft tissue. However, more and more research is adopting a multiphasic structural scaffold with multiple chambers [28,49,86,93,94,95] in periodontal regeneration, aiming for the entire periodontal regeneration of C, P, and AB [96]. Most of the research claimed to regenerate entire periodontal tissue with harmonious tissue layers adopted a multi-phrasing scaffold combining bioactive molecules and physical signals [28,31,86,93,94]. Moreover, combinations of extracellular components such as dECM, exosomes, apoptotic extracellular vesicles, and stem cells were also of great potential since this therapy refill the defect area with more of the same components as the original tissue in the first place, and dECM could function as a stem cell carrier as well as promote cell-ECM communication through activation of the signaling pathway for transduction [32].

With the development of soluble dECM, 3D printing and other advanced fabrication techniques have helped improve the fidelity and strength of dECM-based 3D scaffolds [97], focusing on the spatial management in the defect area [87,98,99]. The preservation of tissue niches provides a relatively independent microenvironment interconnected with tissue engineering constructs with an ideal degradation rate. Moreover, many scientists also focus on guiding periodontal ligament fiber attachment from the root surface to the alveolar bone to generate periodontium with the function of anchoring the tooth in the bone as well as the transduction of occlusive forces [95,100].

To sum up (Table 3), given the complexity of periodontal tissue, direct application of one type of dECM may not have the full potential to regenerate the entire periodontal tissue due to the lack of regenerative niches for multilayer tissue as well as the lack of tailored signals for specific tissue. Therefore, choices should be carefully considered when adopting dECM for various purposes.

## 4. Summary and Prospects

It has been shown that dECM is a complex material composed of a variety of key components associated with regeneration that can regulate cellular responses and thereby activate the body’s process of tissue repair. The combination of dECM materials and tissue engineering methods plays an important role in the treatment of periodontal defects regeneration caused by periodontal diseases. The fabrication and in vitro analysis of dECM (solid dECM and soluble dECM) of various tissue origins are prerequisites for applications. The dECM can be generated and improved by a variety of methods, and complex periodontal regeneration modules based on the dECM were recently developed [20,48,55,87].

In vitro and in vivo studies on the treatment of periodontal defects have shown that both solid and injectable dECM therapies have a regenerative effect, and dECM materials such as decellularized bone powder (Bio-Oss, Geistlich™) have entered clinical use for the treatment of human periodontal diseases. Various other therapies using dECM materials and their derivative materials, stem cells, and/or cytokines also attract great attention. Periodontal treatment based on dECM is not limited to solid state dECM as periodontal guide tissue regeneration (membrane) material, but more focuses on the role of solid or soluble dECM as a potential tissue engineering scaffold to simulate the in vivo microenvironment, determine stem cell fate, and promote tissue regeneration. Over the past decade, as dECM hydrogels are gradually developed, more and more studies have overcome defects in nature’s dECM physical properties, clarified the biological effects of dECM, and are going to guide the next generation of periodontal regeneration.

Complete periodontal tissue regeneration was the ultimate purpose of periodontal regeneration. However, there remain challenges in achieving complete periodontal regeneration with dECM application due to the shrinkage, distortion, collapse, and lack of strength of dECM [22]. The preparation of soluble dECM by adjusting pH, salt or ion concentration, and temperature of the dECM solution can promote gelation only to a certain extent and cannot significantly improve the mechanical properties and biological behavior of dECM. Therefore, functional groups are crosslinked by physical and chemical methods and will make a revolution in the future fabrication and application of dECM by improving the mechanical properties of dECMs and promoting the repair of damaged tissues. Meanwhile, since dECM materials are abundant in certain functional groups such as hydroxyl, acyl, peptide bonds, and hydrogen bonds, they promote drug loading through crosslinking reactions such as amide crosslinking [104] and crosslinks via the interaction of aldehyde groups with amine groups of lysine or hydroxylysine [45]. During the past twenty years, scientists have attempted to modify dECM material to improve the biomechanical performance and maximize the translational potential of dECM. However, there are still challenges in providing dECM into certain shapes with an ideal microstructure, especially in the application of periodontal tissue regeneration, where the defect could vary from one individual to another.

Recently, a large number of scientists focused on the spatiotemporal stability of degradable scaffolds with controlled degradation speeds, which was of great interest [1,105]. Moreover, more and more scientists have been trying to adopt new technology, such as electrospinning, aiming for the fabrication of functionally graded scaffolds [93,95,106,107]. Multi-phasic scaffolds were proved to provide favorable compartmentalization so that progenitor cells from the resident periodontal tissues were capable of regenerating the various periodontal compartments, according to the principles of guided tissue regeneration. Since the development of 3D bioprinting techniques and soluble dECM bioinks, researchers are able to fabricate defect-tailored bioprinted dECM patches for personal periodontal regeneration. Moreover, tissue-specific dECM could be used in preserving tissue-specific niches and could provide a multi-chamber bioscaffold with tissue-specific niches in independent chambers, which has great potential for preserving “tissue-niches” with different bioactive factors [1,105]. In addition, an integrated biodegradable scaffold with independent chambers for different tissue development in the periodontal defect was critical for periodontal tissue regeneration, both in terms of regenerating complicated tissue and the integration of the interfaces of different structures. Therefore, a dECM-based multifunction scaffold with ideal structure could not only serve as a scaffold but also function as a bioactive factor that acted as an initiator in periodontal tissue engineering.

However, although the past few decades have witnessed substantial progress in dECM developments, there remain challenges in understanding the accurate and specific key ECM proteins and the ratio of these molecules for cell proliferation and targeted cell differentiation for 3D organoid culture and tissue repair. Thus, further basic research and preclinical testing are necessary before clinical translation to better understand how the network system exerts a positive effect on stem cells and cell-ECM interaction.

Moreover, given that ECM is an essential part of body tissue, cell/tissue-derived dECM preserving its structures and properties is critical for developing 3D microenvironments for tissue regeneration models for cell therapy or cell-free therapy in periodontal regeneration. Macromolecular materials such as collagen, Matrigel, and electrospun polymer membranes mimicking natural ECM structure are generic products widely used in periodontal tissue engineering. However, tissue engineering constructs for complicated periodontal tissues (cementum, periodontium, or alveolar bone) need to be developed, and specialized dECM is required to be tailored for either C, P, or AB. Since different tissues have different preferences for porosity, permeability, mechanical properties, and, more importantly, biological cues of different dECMs, it is necessary to generalize the natural origin of the ECM of specific tissues of periodontal tissue or tailor an ECM with a biological effect on specific stem cells that proved to play a key role in promoting relative tissue (C, P, AB) regeneration.

## Figures and Tables

**Figure 2 biomolecules-13-00673-f002:**
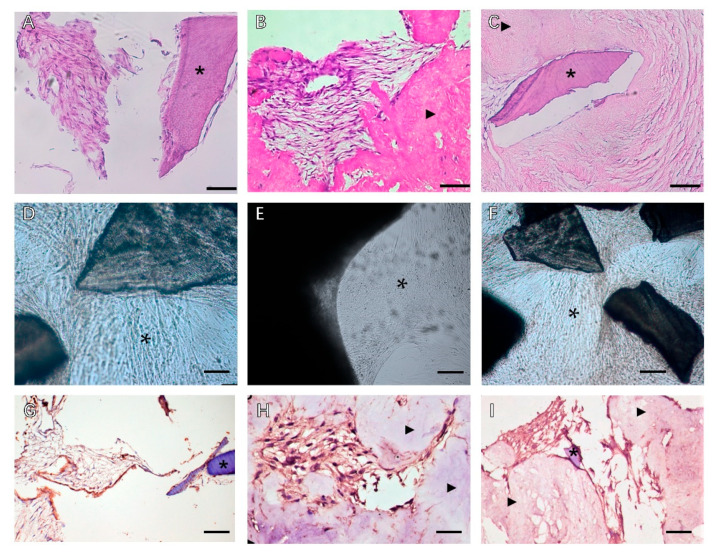
Phase contrast microscopy of decellularized scaffolds cultured with periodontal cells in 3D culture for 14 days. Scale bars, 200 µm in (**A**–**C**): (**A**) PDLSCs around the T-dECM; (**B**) periosteum stem cells (PSCs) around and in the dPDL; (**C**) PDLSCs + PSCs around the T-dECM and in the dPDL. The asterisk indicates the monolayer of the cells. Histological sections of decellularized scaffolds were cultured with periodontal cells in 3D culture for 14 days. Hematoxylin-eosin staining. Scale bars, 100 µm in (**D**), 50 µm in (**E**,**F**): *: PDLSCs next to the dTM indicated by an asterisk; (**E**) PSCs in the dPDL (**F**) PDLSCs + PSCs around the T-dECM and in the dPDL (**D**) original magnification ×200; (**E**,**F**) original magnification ×100. Evaluation of the expression of osteogenic differentiation markers in periodontal cells cultured on decellularized scaffolds under 3D culture for 14 days: (**G**–**I**) Immunohistochemical staining. The positive cells have a brown color. The nuclei were counterstained with hematoxylin. Scale bars, 100 µm in (**G**–**I**): (**G**) OPN staining of PDLSCs next to the T-dECM indicated by an asterisk; (**H**) OC staining of PSCs in the dPDL indicated by arrowheads; (**I**) OPN staining of PDLSCs + PSCs around the T-dECM and in the dPDL indicated by an asterisk and arrowheads, respectively; Reprinted from Ref. [29]. *: MSCs around T-dECM(dTM); 
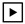
: MSCs around T-dECM(dPDL).

**Figure 3 biomolecules-13-00673-f003:**
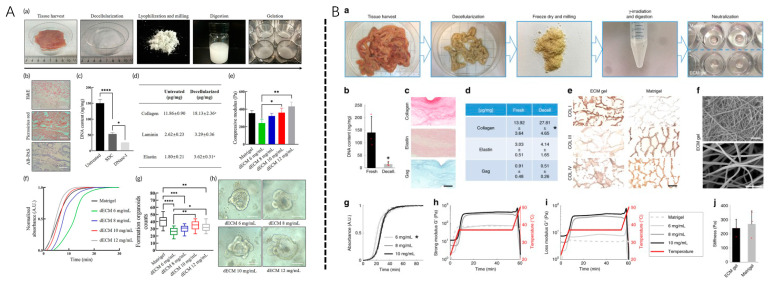
Decellularized extracellular matrix-based (**A**) bio-inks and (**B**) hydrogels. (**A**) Characterization of fabricated decellularized extracellular matrix-based bio-inks. Adapted from Ref. [53]. (**B**) Fabrication and characterization of ECM hydrogels. Adapted from Ref. [54]; (**Aa**) Composite bio-inks of different concentrations of decellularized extracellular matrix (dECM) powder or gelatin methacrylate (GelMA) for organoid culture and bioprinting. (**Ab**) SEM images of dECM-inks with porous microstructures after gelation, * *p* < 0.05. (**Ac**) Frequency-sweep; * *p* < 0.05 and **** *p* < 0.0001. (**Ad**) Strain-sweep; and (**Ae**) Time-sweep of dECM-inks upon blue light irradiation. * *p* < 0.05 and ** *p* < 0.01. (**Af**) Viscosity with shear rates of different dECM-inks. (**Ag**) Elastic modulus measured by compressive testing. One-way ANOVA. * *p* < 0.05, ** *p* < 0.01, *** *p* < 0.001, and **** *p* < 0.0001. (**Ah**) Microscope images of printed lattice patterns. (**B**): Extracellular matrix hydrogel characterization. (**B**) The gelation preparation protocol. (**Bb**) DNA quantification in fresh and decellularized piglet mucosa. (**Bc**) Histological sections of fixed ECM gel drops. (**Bd**) Quantification of ECM proteins: collagen, elastin, GAG, and collagen types, * *p* < 0.05 (**Be**) in ECM gel and Matrigel (**Bf**) Scanning electron microscopy (SEM) images; and (**Bg**) Spectrophotometry of the ECM gel. * *p* < 0.05. (**Bh**,**Bi**) Oscillatory rheology provides a rheological profile of various concentrations of the ECM gel and Matrigel; (**Bj**) Elastic modulus was measured by nanoindentation of 6 mg/mL ECM gel vs. Matrigel in 30 µL drops. Red dots show individual data points.

**Table 3 biomolecules-13-00673-t003:** Application of soluble dECM in periodontal regeneration.

Soluble	Soluble Degree	Cell/Cytokine/Scaffold	Model	Application
Particle/powder [8,70,71]	partial	Hydrogel/guiding tissue membrane [50,71]	Rat periodontal defect model; Calvarial defect model; subdermal implantation	Suspension [69] and scaffold loading [67,68] Bone powder [101,102,103].
Nanoparticle/nanodrops/	partial	-	In vivo; In vitro	Combined with electrojet technology or phase separation technology [73].
2D coating	completely	PDLSC	In vitro	Assessment of cellular responses to different dECM allows for parameters and evaluation of high-throughput analysis of rare or inaccessible pathological dECM in multiple cells [77].
2D hydrogel	completely	PDLSCs/DFCs	In vitro	With biochemical features of a tissue-specific extracellular matrix that guide the environment of cell growth [54].
3D hydrogel	completely	cytokines	Rat periodontal defect model; Cranial defect model; subdermal implantation	3D printing technology is a useful method; multilayer dECM bioprinting scaffolds and multilayer hydrogel systems with spatial segmentation can provide controllable release of multiple growth factors to induce orderly regeneration of periodontal defects [85].
Bio-inks	completely	DFCs+dECM+GelMA	Rat periodontal defect model +Beagle dog periodontal defect	With potent dECM immunomodulatory activity, ECM reduces proinflammatory factors released by M1 macrophages and reduces local inflammation in periodontal defects in SD rats; enhanced regeneration of beagle dog periodontal tissue, especially forming the anchor structure of the ideal bone-periodontal ligament interface, well-arranged periodontal fibers, and highly mineralized alveolar bone [87].

## Data Availability

No new data were created.

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
