# Peer review of "Advances Focusing on the Application of Decellularized Extracellular Matrix in Periodontal Regeneration"

_biomolecules, 2023, doi:10.3390/biom13040673_

Round 1

Reviewer 1 Report

Advances focusing on application of decellularized matrix in periodontal regeneration”.

While this review summarizes well the recent documents, it needs some major changes as commented below:

1. Significant deficit of this review is lack of figures. A few figures will greatly help.

2. Preservation of tissue niche of decell matrix is very important, so please refer to the below refs and discuss this point also in the review:

- Preservation of the naïve features of mesenchymal stromal cells in vitro: Comparison of cell- and bone-derived decellularized extracellular matrix. 2022.

- Decellularization for the retention of tissue niches. 2022.

3. Pls explain the pro and cons of cell-dependent decell matrix

4. Please explain other tech. advances for periodontal TE, with citing some key references below:

- Periodontal Wound Healing and Tissue Regeneration: A Narrative Review, 2021.

- Recent update on craniofacial tissue engineering. 2021.

5. Future trends, challenging aspects should be included and discussed. 

Reviewer 2 Report

Dear authors,

I read carefully your MS about regeneration of periodontal dECM based. It is well written, with precision, and all terms are well explained. It is a good state of art of this topic.

I have just a question. Why did you not include this reference in your review paper:

Ivanov, A. A., Danilova, T. I., Kuznetsova, A. V., Popova, O. P., & Yanushevich, O. O. (2023). Decellularized Matrix Induced Spontaneous Odontogenic and Osteogenic Differentiation in Periodontal Cells. Biomolecules, 13(1), 122.

The MS needs a carefull proofreading, to correct some mistake, as “3.2 ssoluble”.

I endorse publication.

Regards,

Reviewer 3 Report

Please see the word file attached.

Round 2

Reviewer 3 Report

Please see an attached file.
